# Development of a Clinical Score to Stratify the Risk for Carbapenem-Resistant Enterobacterales Bacteremia in Patients with Cancer and Hematopoietic Stem Cell Transplantation

**DOI:** 10.3390/antibiotics12020226

**Published:** 2023-01-20

**Authors:** Fabián Herrera, Diego Torres, Ana Laborde, Lorena Berruezo, Rosana Jordán, Inés Roccia Rossi, Alejandra Valledor, Patricia Costantini, Miguel Dictar, Andrea Nenna, María Laura Pereyra, Sandra Lambert, José Benso, Fernando Poletta, María Luz Gonzalez Ibañez, Nadia Baldoni, María José Eusebio, Fiorella Lovano, Laura Barcán, Martín Luck, Agustina Racioppi, Lucas Tula, Fernando Pasterán, Alejandra Corso, Melina Rapoport, Federico Nicola, María Cristina García Damiano, Ruth Carbone, Renata Monge, Mariana Reynaldi, Graciela Greco, Marcelo Bronzi, Sandra Valle, María Laura Chaves, Viviana Vilches, Miriam Blanco, Alberto Ángel Carena

**Affiliations:** 1Centro de Educación Médica e Investigaciones Clínicas (CEMIC), Buenos Aires C1431, Argentina; 2FUNDALEU, Buenos Aires C1114, Argentina; 3Hospital HIGA Rodolfo Rossi, La Plata B1902, Argentina; 4Hospital Británico de Buenos Aires, Buenos Aires C1280, Argentina; 5Hospital HIGA Gral San Martín, La Plata B1900, Argentina; 6Hospital Italiano de Buenos Aires, Buenos Aires C1199, Argentina; 7Instituto de Oncología Angel H. Roffo, Buenos Aires C1417, Argentina; 8Instituto Alexander Fleming, Buenos Aires C1426, Argentina; 9Hospital Municipal de Oncología Marie Curie, Buenos Aires C1405, Argentina; 10Hospital Universitario Austral, Buenos Aires B1629, Argentina; 11Hospital El Cruce, Buenos Aires B1888, Argentina; 12Hospital Italiano de San Justo, Buenos Aires C1198, Argentina; 13Servicio de Antimicrobianos, ANLIS Malbrán, Buenos Aires C1282, Argentina

**Keywords:** score, Enterobacterales, bacteremia, cancer

## Abstract

Identifying the risk factors for carbapenem-resistant Enterobacterales (CRE) bacteremia in cancer and hematopoietic stem cell transplantation (HSCT) patients would allow earlier initiation of an appropriate empirical antibiotic treatment. This is a prospective multicenter observational study in patients from 12 centers in Argentina, who presented with cancer or hematopoietic stem-cell transplant and developed Enterobacterales bacteremia. A multiple logistic regression model identified risk factors for CRE bacteremia, and a score was developed according to the regression coefficient. This was validated by the bootstrap resampling technique. Four hundred and forty-three patients with Enterobacterales bacteremia were included: 59 with CRE and 384 with carbapenem-susceptible Enterobacterales (CSE). The risk factors that were identified and the points assigned to each of them were: ≥10 days of hospitalization until bacteremia: OR 4.03, 95% CI 1.88–8.66 (2 points); previous antibiotics > 7 days: OR 4.65, 95% CI 2.29–9.46 (2 points); current colonization with KPC-carbapenemase-producing Enterobacterales: 33.08, 95% CI 11.74–93.25 (5 points). With a cut-off of 7 points, a sensitivity of 35.59%, specificity of 98.43%, PPV of 77.7%, and NPV of 90.9% were obtained. The overall performance of the score was satisfactory (AUROC of 0.85, 95% CI 0.80–0.91). Finally, the post-test probability of CRE occurrence in patients with none of the risk factors was 1.9%, which would virtually rule out the presence of CRE bacteremia.

## 1. Introduction

Over the last two decades, there has been a worldwide emergence of CRE, which has become one of the most challenging threats to the healthcare-system. The most common resistance mechanism is the presence of a carbapenemase gene, which is mostly found in *Klebsiella pneumoniae*. However, the genes encoding for carbapenemases are typically located on plasmids and can be transferred both within bacterial species and across different species and genera. These plasmids often coharbor resistance mechanisms against other antimicrobial classes, such as fluoroquinolones, aminoglycosides, tetracyclines, and trimethoprim-sulfamethoxazole, rendering bacteria resistant to multiple drugs [1,2]. From 1997 to 2016, the SENTRY Antimicrobial Surveillance Program collected Enterobacterales isolates from 42 countries located in four main geographic regions. Statistically significant increases in CRE rates have been observed worldwide, with the highest rates reported in Latin America. Although KPC-encoding genes were the most prevalent in several geographic areas, Enterobacterales species that carry genes encoding NDM and OXA-48 variants are on the rise. An increase has also been observed in Enterobacterales isolates that harbor ≥2 carbapenemase genes [3,4,5]. This problematic scenario is a challenge to antimicrobial treatment because of the limited therapeutic options evidenced by the poor outcomes of carbapenemase-producing Enterobacterales (CPE) infections [6].

The epidemiology of multidrug-resistant gram-negative bacteria (MDR-GNB) in cancer and HSCT may differ according to the pathogen, geographic region, country and among health-care centers in the same country.

In this context, an intercontinental study on bacteremias due to GNB in HSCT in 65 centers from 25 countries provided important data. CRE was found in 8.4% of the total Enterobacterales, with 25% being *K. pneumoniae* isolates. Regarding the resistance to carbapenems, a substantial difference was observed among the regions of Europe (20.7% in the southeast vs. 4.9% in the northwest, *p* < 0.0001) [7]. Likewise, a multicenter study conducted in 9 Italian hospitals found that among *K. pneumoniae* strains, 34.9% were resistant to carbapenems [8]. The Argentine Group for the Study of Bacteremia in Cancer and Stem Cell Transplant (ROCAS) registered 1277 bacteremia episodes in patients with hematologic malignancies (HM) and HSCT, and 60% of the isolates were GNB (75% Enterobacterales). The resistance to meropenem in patients with HM and HSCT was 18.4% vs. 26.4% (*p* = 0.016), which was mainly mediated by KPC-carbapenemase-producing Enterobacterales (KPC-CPE) [9].

The overall 30-day mortality rates after CRE bacteremia in patients with HM and HSCT range from 50% to 72.7% [10,11,12,13]. A large study on Enterobacterales bacteremia in cancer and HSCT patients carried out in Argentina found that CRE bacteremia was independently associated with higher 30-day mortality (OR: 7.4; 95% CI: 3.4–15.9; *p* < 0.001) [14].

Several risk factors for CRE bacteremia have been identified in hematological patients. They include rectal colonization by CRE, recent use of carbapenems and other antibiotics, intensive care unit (ICU) admission, acute myeloid leukemia, glucocorticoids, longer stay of current hospitalization before the episode, etc. [15,16,17,18]. However, some of the abovementioned risk factors have not been precisely defined in the literature, such as length of hospital stay and rectal colonization before bacteremia, or the duration of antibiotic treatment until bacteremia. Therefore, we developed and validated a clinical score to accurately define the risk of cancer and HSCT patients developing CRE bacteremia.

## 2. Results

A total of 443 patients with Enterobacterales bacteremia were included: 266 (60%) had HM (with acute leukemia, 39.5%, and lymphoma, 25.5%, being the most frequent); 92 (20.8%) had solid tumors (ST); and 85 (19.2%) had undergone HSCT (42.4% allogeneic). Recently diagnosed was the most frequent stage of underlying cancer (35.9%), and 310 (70%) patients had neutropenia, 87.4% were classified as high-risk by their MASCC score, with a median neutropenia duration of 12 days. Two hundred and thirty-three (53.3%) patients had been hospitalized recently (1 month prior to bacteremia) and 203 (45.8%) had recently used antibiotics. The baseline differences between patients with CSE and CRE bacteremia are detailed in Table 1.

Most isolates were *E. coli* (204, 46%) and *Klebsiella* spp. (182, 41.1%), and 171 (38.6%) microorganisms were MDR, with extended-spectrum β-lactamases (ESBL)-producing Enterobacterales (108, 24.4%) and KPC-CPE (54, 12.2%) being the most frequent. No outbreaks due to MDR-GNB were observed during the study period. The microbiological characteristics of CSE and CRE bacteremia are described in Figure 1 and Figure 2.

Two hundred and ninety-six (66.8%) of the bacteremia cases were classified as nosocomial infections, and the length of hospitalization until bacteremia was 9 days (interquartile range (IQR): 0–15). Three hundred and twenty (72.2%) bacteremia cases had a clinical source, with abdominal being the most frequent (129, 29.1%). The empirical antibiotic treatment (EAT) was adequate in 85.8% of the patients, with monotherapy (321, 72.4%) being the most frequent. One hundred and thirteen patients (25.5%) developed shock, and 34 (7.7%) had breakthrough bacteremia. The 7-day and 30-day mortality was 14.9% and 22.6%, respectively, and was related to infection in more than 70% of the cases. Marked differences were observed in the treatment and outcomes between the two groups, as highlighted in Table 2.

Univariate analyses identified eight risk factors for CRE bacteremia, and multivariate analysis showed that three were independently associated. To predict CRE bacteremia, a score was developed by assigning a point to each of these variables according to their regression coefficient, as shown in Table 3.

The sensitivity, specificity, likelihood ratios (LR) and post-test probability of CRE bacteremia for each cut-off point of the score are shown in Table 4.

No collinearity was observed between these variables. Using a cut-off value of 7 points, the score had a sensitivity of 35.59% (95% CI 26.3–49.1%), a specificity of 98.43% (95% CI 96.6–99.4%), a positive predictive value (PPV) of 77.7% (95% CI 59.5%–89.2%), and a negative predictive value (NPV) of 90.9% (95% CI 89.2%–92.3%). Internal validation was performed using the bootstrap resampling technique (1000 samples). We calculated the possible deviation of ß-coefficients (bias) with their respective confidence intervals, and the results were compared in regard to both calibration and discrimination, with a very high similarity between them. This final model reported an area under the ROC curve (AUROC) of 0.85; 95% CI 0.79–0.90 and an AUROC of 0.85; 95% CI 0.80–0.91 for the internal validation. In the cases where none of the risk factors was identified, the negative LR was 0.12 and the post-test probability of presenting CRE bacteremia was 1.90%.

## 3. Discussion

Several score systems have been developed to more precisely define the risk for CRE infection and improve the clinical approach to reduce mortality, and to further avoid the unnecessary use of broad-spectrum antibacterial agents. In this regard, Gianella et al. designed a score to identify the risk factors for CR-*K. pneumoniae* bacteremia among rectal carriers in a multicenter prospective matched case-control study with 143 bacteremia episodes and 572 controls without infection. Admission to the ICU, abdominal invasive procedures, chemotherapy/radiation therapy, and the number of additional colonization sites were independent risk factors for CR-*K. pneumoniae* bacteremia. Since the score had a high NPV, they concluded that anti-CR-*K. pneumoniae* antibiotic coverage could be avoided in rectal carriers without the above risk factors, and who develop signs and symptoms of infection [19]. A few years later, Cano et al. validated this score in a cohort of 94 CR-*K. pneumoniae* rectal carriers [20]. Although the first study included immunosuppressed patients, the proportion with malignancies and neutropenia was low. The second study was carried out during a protracted outbreak of infections due to clonal KPC-*K. pneumoniae*, and bacteremia episodes only accounted for 50% of them. Since our study markedly differs in terms of the patient population, epidemiological characteristics, and risk factors, the results cannot be extrapolated to those of the previously described studies.

Gianella et al. conducted a study in 237 liver transplant patients screened for rectal CR-*K. pneumoniae*. The independent risk factors for CR-*K. pneumoniae* infection were renal replacement therapy, mechanical ventilation > 48 h, hepatitis C virus recurrence, and colonization with CR-*K. pneumoniae* at any time was the strongest variable [21]. As observed in the previous studies, the population’s characteristics differed from ours. In addition, although colonization was the risk factor most strongly associated with infection, this variable was not differentiated in terms of the time of occurrence. We found that only recent colonization with KPC-CPE was significantly associated with the risk of bacteremia.

Three retrospective single-center studies have been published, which propose risk prediction models for CRE infections and bacteremia in patients with HM and HSCT. Zhang et al. [22] performed a study in 734 patients (22 with CR-*K. pneumoniae* bacteremia). They did not identify the resistance mechanism to carbapenems (e.g., enzyme type such as KPC or OXA carbapenemases). The following three risk factors for CR-*K. pneumoniae* were identified: invasive mechanical ventilation, CR-*K. pneumoniae* rectal colonization and severe neutropenia. The risk score was considered adequate for the prediction of CR-*K. pneumoniae* bacteremia, with an AUROC of 0.753 (95% CI = 0.648–0.858). Nevertheless, the actual predictive performance of the model including sensitivity, specificity, PPV and NPV was not shown.

Wang et al. [23] conducted a 1:1 matched case-control study on intestinal carriers hospitalized in the HM unit and ICU who developed CRE bacteremia and compared them with those who did not develop CRE infection. A total of 42 cases were included. Gastrointestinal injury, tigecycline exposure and the carbapenem resistance score were independent risk factors for CRE bacteremia. The cut-off value of the model was 0.722, with the sensitivity, specificity and AUROC being 90.5%, 85.7% and 0.921, respectively. However, the number of cases was small and the model was developed in both the hematology and ICU department, which could lead to different epidemiology and risk factors for infection.

Wu et al. [24] performed a study on 392 patients with HM who were colonized with CR-GNB (Enterobacterales and non-fermenters GNB). Of these, 10.7% developed a confirmed CR-GNB infection (10.2% were bacteremia), and 36% developed a clinically diagnosed infection (presumed by their therapeutic response to antibiotics). However, some or several of these infection episodes could not be due to CR-GNB, and the risk was therefore overestimated. Moreover, infections caused by both Enterobacterales and non-fermenters CR-GNB were included, with a different risk of infection regarding intestinal colonization.

Our study differs from the three described above because it is prospective and multicentric, has a larger population of immunosuppressed patients, all with Enterobacterales bacteremia and with the identification of the CRE mechanism.

Identifying patients at risk for CRE has a significant clinical impact since adequate empirical treatments can be implemented early, thus reducing mortality. In our cohort, patients with CRE bacteremia compared with those with CSE bacteremia had significantly less adequate empirical treatment, more significant delay in initiating adequate antibiotic treatment before 24 h, and higher 7-day mortality and 30-day mortality. In this regard, several studies on CRE infections have shown that initial inadequate antimicrobial therapy was significantly associated with higher mortality, while adequate early antimicrobial therapy (administered within 24 h from blood culture collection) improved survival [13,15,25].

No randomized clinical trials have been completed to define the optimal treatment for CRE infections. The existing clinical evidence regarding treatment is based upon observational clinical data, in vitro data, and in vivo animal models. In recent years, new drugs have become available for the treatment of carbapenem-resistant infections, including, among others, ceftazidime–avibactam, meropenem–vaborbactam and imipenem–relebactam. The European Society of Clinical Microbiology and Infectious Diseases (ESCMID) guidelines recommend ceftazidime–avibactam or meropenem–vaborbactam for severe CRE infections if active in vitro [26]. On the other hand, the Infectious Diseases Society of America guidelines recommend ceftazidime–avibactam, meropenem–vaborbactam and imipenem–relebactam as the preferred treatment options for KPC-CRE infections outside of the urinary tract. However, empiric treatment recommendations are not provided [27]. Ceftazidime–avibactam provided the most extensive real-life data, and demonstrated superiority over other antibiotics for KPC-CRE infections. The larger studies included patients with HM, HSCT, and neutropenia, but they failed to analyze them separately from the total cohort [28,29,30]. A few studies that were exclusively focused on patients with HM have shown clinical benefits over other antibiotics [31,32].

There are encouraging data on treatment with the new ß-lactam/ß -lactamase inhibitor antibiotics, which highlights the early identification of patients with CRE infections.

We acknowledge several limitations of the present study. First, the population analyzed is heterogeneous in terms of cancer type and the presence of neutropenia. However, bloodstream infections in cancer patients may occur in those with even a minor degree of immunosuppression. Second, the data on previous or recent colonization with KPC-CPE may be underestimated because rectal swabs were collected at 11 centers. However, the center that did not perform any surveillance only represented 4.1% of the total cohort, which would therefore not change the interpretation of the results. Third, our score was developed in a cohort where the prevalent mechanism of carbapenem resistance was KPC production. We must determine whether it could also be applied to patients colonized with other types of carbapenemase-producing strains (i.e., MBL, OXA-48). Fourth, this score was developed in a population with a high prevalence of CRE. Therefore, the score could not be applied to areas where cancer patients had different epidemiology.

The strengths of our study rely on its being the first prospective, multicenter study carried out in cancer and HSCT patients, where a large number of Enterobacterales bacteremia episodes have been included. Risk factors for CRE were identified, and a risk score was developed and internally validated. As in our case, several studies have identified rectal colonization with KPC-CPE, recent use of antibiotics and a longer stay of current hospitalization before the episode of bacteremia as risk factors for CRE in different populations. However, our study was able to identify these variables more accurately according to the number of days (>7 days of antibiotics use prior to bacteremia and ≥10 days of hospitalization prior to bacteremia), and the time of occurrence of these risk factors (recent colonization with KPC-CPE). Patients with a cut-off value of 7 points had a specificity of 98.3%, a PPV of 77.7%, and an NPV of 90.9% for presenting CRE bacteremia, which is considered as high risk. However, the most relevant issue is that in patients with none of these risk factors, the negative LR was 0.12 and the post-test probability of presenting CRE bacteremia was 1.90%. Therefore, it could be useful to rule out CRE bacteremia, especially in regions with a lower CRE prevalence.

Another important point is that two of the identified risk factors may be modified with appropriate infection control measures and antimicrobial stewardship programs [33,34]. Likewise, the strategy of fecal microbiota transplantation for decolonization of patients with CRE is also promising [35,36].

To conclude, this risk score in cancer and HSCT patients could help avoid the use of broad-spectrum antibiotics in patients at a low risk of developing CRE bacteremia and help to prescribe an early optimal antibiotic regimen in high-risk patients in order to improve the outcome. Nevertheless, to clearly estimate the utility of this score in clinical practice, it must be validated in a prospective cohort.

## 4. Materials and Methods

### 4.1. Setting, Patients and Study Design

A multicenter prospective study was performed in 12 referral teaching centers (7 private and 5 public) that specialize in the care of oncological and transplant patients in Argentina. The patient characteristics and the medical treatment provided were similar in these centers.

All episodes of initial monomicrobial bacteremia (defined as the first episode of bacteremia experienced during an admission) caused by Enterobacterales in adult patients (≥18 years of age) that were managed as inpatients from May 2014 to June 2019 were included. Patients presented with (a) a solid tumor or HM treated with chemotherapy or biological agents (six months prior to admission) or they had been receiving steroids (at a dose equal to or higher than prednisone 20 mg daily or equivalent, for at least two weeks prior to admission); or (b) allogeneic HSCT (with graft versus host disease at any time or without this disease in the first two years) or autologous HSCT (in the first year post-transplant). Patients with polymicrobial bacteremia, and those receiving palliative care were excluded from the analysis. Patients were identified at the time of a positive blood culture and were then followed prospectively. Data were obtained from medical records (electronic and paper records depending on the center involved) and direct patient care, with a double check made with microbiological records from the laboratory. Clinical, microbiological, treatment, and outcome variables were evaluated. Empirical personalized antibiotic therapy was started based on the patient’s clinical and epidemiological features, and at the discretion of the attending physician. Patients were followed for 30 days after the episode (by direct patient care in hospitalized cases, or by phone calls in the case of discharged patients), or until the patient’s death, provided that it happened before (assessed by direct patient care in patients still hospitalized or by a local healthcare database in each center). The stage of the underlying cancer and immunosuppression in ST, HM and HSCT patients are outlined in A1, Appendix A.

### 4.2. Definitions

Neutropenia was defined as an absolute neutrophil count < 500 cells/mm^3^. High-risk febrile neutropenia was defined according to clinical variables and a Multinational Association for Supportive Care in Cancer (MASCC) score of < 21 [37]. The clinical source of infection was determined based on the isolation of the bacteria in the suspected source and/or the associated clinical signs and symptoms. Recent antibiotic use was defined as any antibiotic administered 30 days before the episode of bacteremia and for more than 48 h. Recent ICU admission was defined as an admission within 14 days prior to the episode of bacteremia and for at least 72 h. A central venous catheter was considered a risk factor when placed for at least 72 h before the episode of bacteremia. Colonization or infection with CRE was defined as “previous” when it occurred within six months before hospitalization, and “recent” when it was detected within one week of the episode of bacteremia.

Friedman et al. classified bacteremia as nosocomial, health-care-associated, or community-acquired [38]. Breakthrough bacteremia was defined as an episode of continuous or new-onset bacteremia in a patient receiving appropriate antibiotics for the microorganism that was recovered from blood cultures. The empirical antibiotic treatment (EAT) was adequate if one or more antibiotics were active in vitro against the isolated bacteria. In patients with ESBL-E, empirical therapy with piperacillin/tazobactam or cefepime alone was considered inadequate [39]. In patients with CSE or CRE, empirical therapy with tigecycline as the only active drug was considered inadequate. Response to treatment on day 7 of therapy was defined as the absence of fever and hypotension, and clinical improvement. Mortality was related to infection provided that there was microbiological, histological, or clinical evidence of active infection.

### 4.3. Microbiological Studies

Bacteremia was defined as the isolation of a pathogenic bacteria in at least one bottle of blood culture (BD BACTEC F Aerobic and Anaerobic, analyzed with BACTEC FX BD, BacTALERT 3D Biomerieux depending on the method available at each center), for a minimum incubation period of five days. MDR-GNB were defined as a GNB resistant to three or more of the following categories of antibiotics: carbapenems, piperacillin/tazobactam, third and fourth generation cephalosporins, aztreonam, fluoroquinolones, or aminoglycosides [40,41]. Microbiological identification and susceptibility testing were done with manual biochemical and microbiological methods, disk diffusion (according to the CLSI recommendations), and/or Etest, VITEK II Compact (bioMerieux), PHOENIX 100 BD (Becton Dickinson), VITEK MS (bioMerieux) and MALDI-TOF (Microflox from Bruker). Carbapenemase production was investigated in carbapenem-resistant bacteria using the modified Hodge method, disk synergy tests with a carbapenem disk placed close to the boronic acid disk test for KPC, and the EDTA disk for identification of metallo-β-lactamases. The presence of genes coding for *bla*KPC and *bla*OXA-48 was investigated by monoplex or multiplex polymerase chain reaction (PCR) using specific primers depending on the method available at each center. Multiplex PCR for *bla*VIM, *bla*NDM, *bla*IMP, *bla*KPC and *bla*OXA-48 was used to investigate 17 isolates at the National Reference Laboratory of Microbiology (ANLIS-Malbrán) [42]. In order to detect colonization with KPC, rectal swabs were routinely collected (once a week and in every pre-transplant evaluation) in 11 of the 12 centers included in the study, using chromogenic methods and/or PCR.

### 4.4. Statistical Analysis

To compare the characteristics, treatments, and outcomes of patients with CRE and CSE, the x^2^ test, Fisher’s exact test and Mann–Whitney U-test were used when indicated. To identify the risk factors for CRE that were included as variables in the score, a multiple logistic regression model (forward-stepwise selection) was used. Variables with a *p* < 0.05 in the univariate analysis were included in the multivariate model. For multivariate analysis, variables with a statistically significant association at *p* < 0.05 were used to develop the score. Weighted scores for each variable were calculated by dividing each regression coefficient by one-half of the smallest coefficient and rounding to the nearest integer [43]. A tetrachoric correlation model was used to evaluate collinearity. The calibration of the model was evaluated using the Hosmer–Lemeshow test, and the Nagelkerke and Cox/Snell R2 were calculated. The predictive performance of the model and the score was evaluated using sensitivity, specificity, PPV, NPV, and AUROC, with a value of 1.0 indicating perfect prediction. The post-test probability of the different values of the score was further assessed for predicting CRE, with a pre-test probability of 13.3% (data obtained from previous local, non-published analysis). The internal validation of the final model was assessed using the bootstrap resampling technique. For all tests, a 95% level of statistical significance was used. All tests were two-tailed. The analyses were performed with the SPSS and Stata 14.0 (Statacorp; College Station, TX, USA) software packages.

## Figures and Tables

**Figure 1 antibiotics-12-00226-f001:**
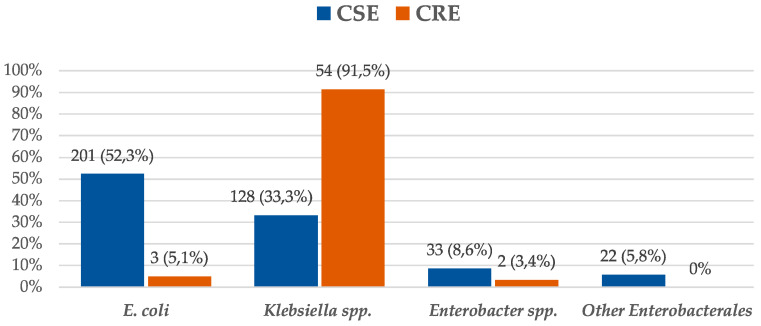
Etiology of Enterobacterales bacteremia. Differences between carbapenem-susceptible Enterobacterales (CSE) and carbapenem-resistant Enterobacterales (CRE) isolates.

**Figure 2 antibiotics-12-00226-f002:**
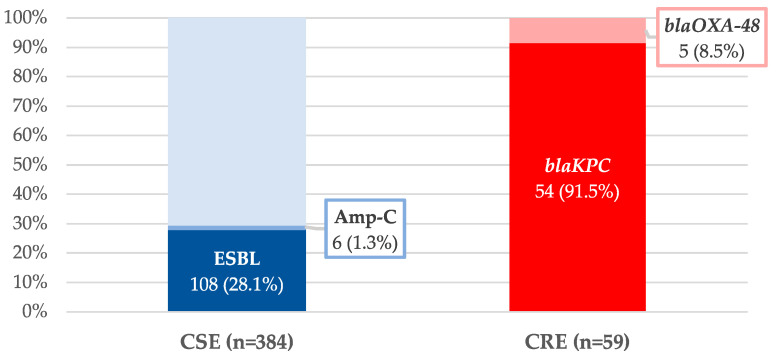
Resistance mechanisms of Enterobacterales. Proportion of extended-spectrum β-lactamases (ESBL) and Amp-C producers in carbapenem-susceptible Enterobacterales (CSE) isolates. Proportion of *bla*KPC and *bla*OXA-48 in carbapenem-resistant Enterobacterales (CRE) isolates.

**Table 1 antibiotics-12-00226-t001:** Baseline characteristics of patients with carbapenem-susceptible Enterobacterales (CSE) and carbapenem-resistant Enterobacterales (CRE) bacteremia.

Variables	CSE *n* = 384 *n* (%)	CRE *n* = 59 *n* (%)	*p*-Value *
Age (years) (median, IQR)	55.5 (37–66)	51 (39–64)	0.31
Male gender	216 (56.3)	35 (59.3)	0.65
Inclusion criteria			
Hematologic malignancy	226 (58.8)	40 (67.8)	0.19
Solid malignancy	89 (23.2)	3 (5.1)	**0.001**
Hematopoietic Stem Cell Transplant	69 (18)	16 (27.1)	0.09
Stage of underlying cancer			
Recently diagnosed	126 (32.8)	33(55.9)	**0.001**
Complete remission	65 (17)	11(18.6)	0.74
Partial remission	57 (14.8)	5 (8.5)	0.23
Refractory	43 (11.2)	3 (5.1)	0.17
Relapse	93 (24.2)	7 (11.9)	**0.03**
Recent chemotherapy (1 month prior to bacteremia)	290 (75.5)	42 (71.2)	0.47
Recent radiotherapy (1 month prior to bacteremia)	28 (7.3)	3 (5.1)	0.78
Receiving steroids during bacteremia	125 (32.5)	35 (59.3)	0.13
Charlson comorbidity index (median, IQR)	2 (2–3)	2 (2–3)	0.21
Neutropenia	258 (67.2)	52 (88.1)	**0.001**
High risk by MASCC score	222 (86)	49 (94.2)	0.10
Neutropenia duration (days) (IQR)	11 (6–19)	16 (10–24)	**0.004**
Recent hospitalization (1 month prior to bacteremia)	204 (53.1)	32 (54.2)	0.87
Recent antibiotic use (1 month prior to bacteremia)	158 (41.1)	45 (76.3)	**<0.001**
Fluoroquinolone prophylaxis	64 (16.7)	13 (22)	0.31

Abbreviation: MASCC, Multinational Association for Supportive Care in Cancer. * *p-*Values obtained by chi-square or Fisher’s exact test for categorical variables, and Mann–Whitney U-test for continuous variables. Bold: statistically significant.

**Table 2 antibiotics-12-00226-t002:** Clinical presentation, treatment, and outcome of carbapenem-susceptible Enterobacterales (CSE) and carbapenem-resistant Enterobacterales (CRE) bacteremia.

Variables	CSE *n* (%)	CRE *n* (%)	*p*-Value *
Nosocomial infection	241 (62.8)	55 (93.2)	**<0.001**
Bacteremia with clinical source	286 (74.5)	34 (57.6)	**0.007**
Abdominal	115 (29.9)	14 (23.7)	0.32
Central venous catheter	40 (10.4)	10 (16.9)	0.14
Urinary tract	55 (14.3)	2 (3.4)	**0.02**
Adequate EAT	352 (91.7)	28 (47.5)	**<0.001**
Empirical antibiotic monotherapy	283 (73.6)	38 (64.4)	0.13
Combined definitive antibiotic treatment	22 (5.7)	57 (96.6)	**<0.001**
APACHE II score the day of bacteremia (median, IQR)	13 (9–17)	12 (8–17)	0.16
PITT score the day of bacteremia (median, IQR)	0 (0–2)	0 (0–2)	0.39
Intensive care unit admission required	86 (22.4)	29 (49.1)	**<0.001**
Shock development	83 (21.6)	30 (50.8)	**<0.001**
Breakthrough bacteremia	20 (5.2)	14 (23.7)	**<0.001**
7-day mortality	45 (11.7)	21 (35.6)	**<0.001**
Infection-related	37 (82.2)	21 (100)	**0.03**
30-day mortality	68 (17.7)	32 (54.2)	**<0.001**
Infection-related	47 (69.1)	27 (84.4)	0.08
More than 24-h delay in adequate antibiotic treatment	27 (7)	26 (44.1)	**<0.001**

Abbreviation: EAT, empirical antibiotic treatment. * *p-*values obtained by chi-square or Fisher’s exact test for categorical variables, and Mann–Whitney U-test for continuous variables. Bold: statistically significant.

**Table 3 antibiotics-12-00226-t003:** Risk factors for carbapenem-resistant bacteremia and points assigned to the score.

Variables	Univariate Analysis		Multivariate Analysis		Points
	OR (95%CI)	*p*-Value	OR (95%CI)	*p*-Value	
Recent carbapenems use	4.1 (2.3–7.7)	0.001			
>7 days of antibiotic use prior to bacteremia	4.6 (2.6–4.1)	0.001	4.65 (2.29–9.46)	<0.001	2
Neutropenia	3.6 (1.6–8.2)	0.002			
≥10-days of hospitalization prior to bacteremia	4.4 (2.3–8.6)	0.001	4.03 (1.88–8.66)	<0.001	2
Recent intensive care unit admission	3 (1.4–6.8)	0.006			
Central venous catheter in place	3.4 (1.7–6.7)	0.002			
Previous colonization with KPC-CPE	10.6 (4.4–25.5)	0.001			
Recent colonization with KPC-CPE	24.9 (1.3–55)	0.001	33.08 (11.7–93.25)	<001	5

Abbreviation: KPC-CPE, KPC-carbapenemase-producing Enterobacterales.

**Table 4 antibiotics-12-00226-t004:** Sensitivity, specificity, and post-test probability of the different cut-off points of the score.

Point of the Score	Sensitivity	Specificity	+LR	−LR	Positive Post-Test Probability	Negative Post-Test Probability
≥0	100.00%	0.00%	1		13.30%	
≥2	94.92%	40.21%	1.5874	0.1265	19.58%	1.90%
≥4	76.27%	83.81%	4.7116	0.2831	41.95%	4.16%
≥5	42.37%	97.13%	14.7535	0.5933	69.36%	8.34%
≥7	35.59%	98.43%	22.7204	0.6543	77.71%	9.12%
≥9	20.34%	99.22%	25.9662	0.8029	79.93%	10.97%
>9	0.00%	100.00%		1	0.00%	13.30%

Abbreviations: +LR, positive likelihood ratio; -LR, negative likelihood ratio.

## Data Availability

Data is available upon request. Contact the corresponding author.

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
