# Peer review of "Development of a Clinical Score to Stratify the Risk for Carbapenem-Resistant Enterobacterales Bacteremia in Patients with Cancer and Hematopoietic Stem Cell Transplantation"

_antibiotics, 2023, doi:10.3390/antibiotics12020226_

Round 1

Reviewer 1 Report

The authors have written an article titled “Development of a Clinical Score to Stratify the Risk for Carbapenem-resistant Enterobacterales Bacteremia in Patients with Cancer and Hematopoietic Stem Cell Transplantation”. The manuscript has been well written and technically studies are designed perfectly. Further, each segment of the study is supported with critical discussion. The manuscript can be accepted after minor spelling and grammar.

Author Response

Point 1: The authors have written an article titled “Development of a Clinical Score to Stratify the Risk for Carbapenem-resistant Enterobacterales Bacteremia in Patients with Cancer and Hematopoietic Stem Cell Transplantation”. The manuscript has been well written and technically studies are designed perfectly. Further, each segment of the study is supported with critical discussion. The manuscript can be accepted after minor spelling and grammar.

Response 1: We thank you for your comments. The manuscript had a new English revision after the changes we made in answer to the other reviewer´s suggestions.

Reviewer 2 Report

Herrera et al present an interesting, thorough study attempting to elucidate the risk factors associated with carbapenem resistant enterobacteria in patients presenting with bacteremia with cancer and HSCT.

The overall quality of the study is good however there are some points that must be addressed.

1.Minor language revisions are encouraged if a native speaker is available. While the overall manuscript is readable, some segments are hard to read or feature redundant sentences. One example is the intro "

A statistically significant 91 increase in CRE rates has been observed over time for the overall isolates. However, a 92 more remarkable increase was observed in Latin America, with a 5.6% CRE increase rate, 93 and carbapenem-resistant K. pneumoniae (CR-KPN) being the main cause of CRE increase. 94 The rise in CR-KPN was statistically significant in all regions but was higher in Latin 95 America." Could be better phrased as "Statistically significnat increases in CRE rates have been observed worldwide, however Latin America has seen the highest increase out of all analyzed regions".   2. The introduction is too long, some segments such as the ones discussing results from prior studies should be better featured in the discussion.  3. Methods should not be the last section, it should go right after the introduction. 4. I have various methodological concerns. Firstly, the apparent patient population could be highly heterogeneous as the inclusion criteria is quite broad, different chemotherapeutic agents have different impacts on the immune system and WBC. This is partially observed in the significantly increased rate of HSCT patients in the resistant cohort. Authors should add to the first table were baseline characteristics are described the chemotherapeutic, immune agnets and steroid agents used along with its frequency. Secondly, authors identified 11 "risk factors" in the "epidemiologic" analysis in table 2 yet only feature 8 in table 4. Thirdly, authors may have a borderline number of patients for such a relatively large multiple regression, and its especially seen on some of the confidence intervals which are wide, (one is 11- ~90). Lastly, while authors performed bootstraping and there are various ways to generate risk factor scores, these usually feature a partition of the dataset into a "training set" and a "testing set" usually at around 80/20. Testing the model against itself. Authors mention using stata for data analysis, if familiar with R they may consider using R with the RMS package to perfrom said analisis which could also facilitate a nomogram. Additionally, given that its a multicenter study, authors should provide a comparison of significant event rates such as mortality, resitance rates and such per center to determine if a specific center is inducing bias or if there is a homogeneous sample. 5. Discussion should be summarized as it is too long.  

Author Response

We thank you for all your comments and suggestions.

Point 1. Minor language revisions are encouraged if a native speaker is available. While the overall manuscript is readable, some segments are hard to read or feature redundant sentences. One example is the intro "

A statistically significant 91 increase in CRE rates has been observed over time for the overall isolates. However, a 92 more remarkable increase was observed in Latin America, with a 5.6% CRE increase rate, 93 and carbapenem-resistant K. pneumoniae (CR-KPN) being the main cause of CRE increase. 94 The rise in CR-KPN was statistically significant in all regions but was higher in Latin 95 America." Could be better phrased as "Statistically significnat increases in CRE rates have been observed worldwide, however Latin America has seen the highest increase out of all analyzed regions".   

 Response 1: The manuscript had a new English revision after the changes we made in answer to the reviewer´s suggestions.

This phrase was changed.

Point 2. The introduction is too long, some segments such as the ones discussing results from prior studies should be better featured in the discussion. 

Response 2.  The introduction was shortened.

Point 3. Methods should not be the last section, it should go right after the introduction

Response 3. The manuscript was written according to the instructions for authors and the template, where methods go in the last section.

Point 4. I have various methodological concerns.

Firstly, the apparent patient population could be highly heterogeneous as the inclusion criteria is quite broad, different chemotherapeutic agents have different impacts on the immune system and WBC. This is partially observed in the significantly increased rate of HSCT patients in the resistant cohort. Authors should add to the first table were baseline characteristics are described the chemotherapeutic, immune agnets and steroid agents used along with its frequency. 

Response 4.

4.1. Although we are aware that the patient population could be heterogeneous, as we described in Methods, all patients herein included were immunosuppressed. When the study was designed, we decided to include the variables associated with high risk for CRE regarding immunosuppression, such as chemotherapy and radiotherapy (Gianella et al, reference 19) and steroids use. We did not include the type and dose of radiotherapy, or the type of chemotherapy, or biological agents used, because we did not find in the literature that those variables were associated with CRE risk. In fact, the rate of those variables was similar between CSE and CRE (table 1). To clarify this issue, a new table was added in appendix A (Table A2), which compares baseline characteristics between patients with HM, HSCT, and ST and further shows that steroid use during bacteremia was similar in the three groups. In contrast, patients with ST received more chemotherapy and radiotherapy than the other groups.

Secondly, authors identified 11 "risk factors" in the "epidemiologic" analysis in table 2 yet only feature 8 in table 4.

Response 4.2. Table 2 describes epidemiological characteristics, many of which are risk factors for CRE. We understand that this can be confusing, so this table was removed, and the variables not identified as risk factors were included in tables 1 and 2.

Thirdly, authors may have a borderline number of patients for such a relatively large multiple regression, and its especially seen on some of the confidence intervals which are wide, (one is 11- ~90). Lastly, while authors performed bootstraping and there are various ways to generate risk factor scores, these usually feature a partition of the dataset into a "training set" and a "testing set" usually at around 80/20. Testing the model against itself. Authors mention using stata for data analysis, if familiar with R they may consider using R with the RMS package to perfrom said analisis which could also facilitate a nomogram. 

Response 4.3.

We have applied a multiple logistic regression (forward-stepwise selection) to select the variables included in the score. The logistic model began with 8 independent variables (full model). Then 5 variables were sequentially removed (P-value > 0.05), resulting in a final model with 3 independent variables: > 7 days of antibiotic use prior to bacteremia; ≥10 days of hospitalization prior to bacteremia; and recent colonization with KPC-CPE). The number of 443 subjects included in the analysis (59 with CRE and 384 with CSE) was higher than the minimum sample size (N) for multiple logistic regressions suggested by Freeman's classical formula of N= 10*(k+1)] (Ref.1), where k is the number of independent variables to be estimated. The sample size was also higher than that suggested by Perduzzi et al. (Ref.2) consisting of ten events of interest (patients with CRE) per estimated independent variable; and even higher than estimate of the minimum number of events of interest suggested for multiple logistic regressions using stepwise selection (Ref.3). Thus, we consider that sample size and power were adequate to converge in a fairly robust model with the selection of 3 variables to the score (logit model: N=443; Log likelihood =-117.5; Wald chi2=54.5; p<0.00001; Pseudo-R2=0.32).

This logistic regression was performed with the sw logit command of Stata 14.0 (Statacorp; USA), which fits a logit model for a binary response by maximum likelihood estimations, as the lrm (logistic regression model) of RMS package running in the R environment, which has been suggested by the reviewer.

Furthermore, the predictive performance of the model and the score was evaluated estimating indicators of sensitivity, specificity, PPV, NPV, and AUROC with their 95%CI. To avoid reducing the sample size and power of each analysis, we applied an internal validation using the bootstrap resampling technique instead of a method of partitioning the sample into two training/testing sets (80/20). However, we agree with the reviewer's suggestion that it would be very interesting to make the analysis using this methodology. Nevertheless, we are planning to validate the score performance in a new prospective cohort.

References:

(1). Freeman, D H., Jr. Applied Categorical Data Analysis (Statistics: A Series of Textbooks and Monographs) ISBN 13: 9780824777524.

(2). Perduzzi P, Concato J, Kemper E, Holford TR, Feinstein AR. A simulation study of the number of events per variable in logistic regression analysis. J Clin Epidemiol 1996; 49: 1373-9. DOI: 10.1016/S0895-4356(96)00236-3.

(3). Steyerberg EW, Eijkemans MJC, Habemma DF: Stepwise selection in small data sets: A simulation study of bias in logistic regression analysis. J Clin Epidemiol 1999; 52: 10: 935-42. DOI: 10.1016/S0895-4356(99)00103-1.

Additionally, given that it’s a multicenter study, authors should provide a comparison of significant event rates such as mortality, resitance rates and such per center to determine if a specific center is inducing bias or if there is a homogeneous sample.

Response 4.4.

Table A1 in Appendix A was added to describe the number of patients, % of CRE, and resistant mechanisms among the 12 centers. The comparison among centers of outcome variables, such as mortality, was not done because it was not the aim of the study.

Discussion should be summarized as it is too long.  

The discussion was shortened.

Reviewer 3 Report

This paper aims at describing risk-factors for carbapenem-resistant enterobacterales bacteremia in a mixed population of cancer (both solid and hematologic malignancies) and hematopoietic stem cell transplanted patients. The authors propose a score to stratify the risk of such occurrences. Although this is an important topic as high mortality rates are indeed witnessed in patients with CRE bacteremia, several issues should be addressed before this work can be seen as suitable for publication. Moreover, the clinical application of the proposed score remains unclear and should therefore be better defined/described by the authors.

Major points

-          The authors should clarify throughout the paper whether their aim is to deal with Carbapenem-Resistant Enterobacterales (CRE) as a whole (as implied by the title) or KPC-carbapenemase-producing Enterobacterales (KPC-CPE), or Carbapenem-Resistant Klebsiella pneumoniae (CR-KPN). For example, Table 1 title states CRE when just below, the authors speak of KPC-CPE and KPC-CPE are only dealt with in Table 2. Was it really only KPC-CPE that were analyzed here ? This kind of mix-up goes on throughout the paper, which is confusing for the reader. Another such example is found in the materials and methods section (line 12, line 460-462) where the weekly screening detection for KPC, and KPC only, is listed (not other CRE). A thorough homogeneisation of the use of CRE, KPC-CPE, KPC according to what was actually evaluated in this work is therefore mandatory.

-          The list of the 12 referral centers for this study should be given as well as a description of their major characteristics, and the number of patients included for each center (especially when one of the centers did not collect weekly rectal swabs for the detection of CRE, some did not perform carbapenemase gene detection by PCR,…). Also, more precisions should be given on the types of solid tumors included in this study.

-          Page 3, line 157: « 87.4% classified as high-risk », high-risk of what ? Please explain.

-          Page 4, lines 163-165: Why were the risk factors recent hospitalization, recent antibiotics use, recent colonization with KPC-CPE, and recent ICU admission not included in Table 1 but kept separate in Table 2 ? These two tables should be fused as one.  Recent and previous colonization with KPC-CPE should also be defined in the table as well as recent antibiotic use. Additionally, the statistical tests used to obtain p-values in these tables should be given in the footnote.

-          Page 5, Figures 1 & 2: the number of strains for each occurrence should be given along with the percentage. Indeed, from the materials & methods section, the reader understands that not all of the strains were screened for blaKPC and blaOXA-48 genes (Page 12, lines 458-460). Precise numbers should therefore be given to ascertain the relevance of these observations.

-          Page 6: Figure 3 is unnecessary for understanding the paper and establishing the proposed score. It should therefore be removed.

-          Page 6, table 3: there is room for improvement for this table. First, it should stand on a single page along with its footnote for a better understanding. Second, the item « clinical source » makes no sense to this reader as well as the statistical analysis held on these values. What do the authors mean with this item? Should it not simply be a category under which « Abdominal » « Central veinous catheter » and « Urinary » are found?

-          Discussion : this section is well researched, with a number of studies addressing a similar problem described with their limitations. However, some sentences are misleading and should be toned down. For example, Page 9, lines 313-4, the authors state «Identifying patients at risk for CRE has a significant clinical impact since adequate empirical treatments can be implemented early, thus reducing mortality ». This statement is contradictory with the fact that recent antibitioc use was identified as a risk factor for CRE bacteremia in this study. In this reader’s point of view, the main interest of a score such as the one established in this work is to avoid unnecessary antibiotic treatment in patients identified as low risk and the emphasis should be put on this item rather that adequate empirical treatment for CRE. Moreover, the most consistent risk factor for CRE bacteremia identified in this work and others is the colonization with CRE (mainly intestinal carriage identified through weekly rectal swabs). In such patients, the possibility to reduce the risk through Fecal Microbial Transplantation (FMT) rather than antibiotic use should therefore also be discussed.

-          Page 11, line 407: « Patients already enrolled in the study…were excluded from the analysis ». This statement is puzzling; please clarify.

Minor points :

-          From page 3, please use K. pneumoniae consistently throughout the manuscript

-          Page 3, line 118 : please define ROCAS in « ROCAS study »

-          Page 3, line 155 : what kind of solid tumors were included and how many of each type ?

-          Pages 5 & 6 : figure titles should be positionned below the figures and not above.

-          Page 5, figure 2 : ESBL abbreviation should also be explained in the caption

-          References : please check the correct use of italics for species names throughout this section

Author Response

We thank you for your comments and suggestions.

Point 1: The authors should clarify throughout the paper whether their aim is to deal with Carbapenem-Resistant Enterobacterales (CRE) as a whole (as implied by the title) or KPC-carbapenemase-producing Enterobacterales (KPC-CPE), or Carbapenem-Resistant Klebsiella pneumoniae (CR-KPN). For example, Table 1 title states CRE when just below, the authors speak of KPC-CPE and KPC-CPE are only dealt with in Table 2. Was it really only KPC-CPE that were analyzed here ? This kind of mix-up goes on throughout the paper, which is confusing for the reader. Another such example is found in the materials and methods section (line 12, line 460-462) where the weekly screening detection for KPC, and KPC only, is listed (not other CRE). A thorough homogeneisation of the use of CRE, KPC-CPE, KPC according to what was actually evaluated in this work is therefore mandatory.

 Response 1: We understand that the definitions regarding the type of microorganisms and the resistant mechanism may differ and could be confusing. In the introduction and discussion we described the literature studies focused on CRE, CRE-CPE, CR-K. pneumoniae and KPC-K.pneunoniae, according to the purpose of the manuscript. Our study was designed in 2014 to identify risk factors for CRE, and we planned to enroll patients with different Enterobacterales species and resistant mechanisms to carbapenems, not only K. pneumoniae or CPE. Finally, at the end of the study, all the isolates were CPE, mostly Klebsiella spp and KPC-producing. Since this is a prospective study designed five years before its conclusion, we found it not suitable to change the aim according to the obtained results. In addition, we refer to CRE and KPC-CPE as a resistant profile and resistance mechanism respectively, in abstract, introduction, results, methods and discussion.

Below table 1 (CSE vs. CRE), we refer to KPC-CPE colonization (not infection) because it was the type of carbapenemase studied in rectal swabs in the participating centers at the beginning. Later, some of them incorporated different methodologies to study other carbapenemases. The same concept applies in table 2. The participating centers that performed surveillance of rectal colonization for an epidemiological purpose studied KPC-EPC because in 2014 more than 90% of the CPE isolates expressed this mechanism in our country. That is the reason why we decided to include this risk factor instead of other resistant mechanisms (studied only in some centers and in some patients), as described in methods.

Point 2. The list of the 12 referral centers for this study should be given as well as a description of their major characteristics, and the number of patients included for each center (especially when one of the centers did not collect weekly rectal swabs for the detection of CRE, some did not perform carbapenemase gene detection by PCR,…). Also, more precisions should be given on the types of solid tumors included in this study.

Response 2. The list of the 12 referral centers is described in affiliation. In Methods we added that all of them are teaching hospitals and mentioned the number of public and private centers. Table A1, Appendix A, shows the number of patients, CRE, and resistant mechanisms included in each center.

Only immunosuppressed patients were included when the study was designed, as described in methods. We didn´t include the type of solid tumor because we didn´t find in the literature that this variable was related with CRE risk. For clarify this subject, a new table was added in appendix A (Table A 2) that compared baseline characteristics between patients with HM, HSCT, and ST and shows that receiving steroid use during bacteremia was similar between the three groups. In contrast, patients with ST received more chemotherapy and radiotherapy than the other groups.

Point 3. Page 3, line 157: « 87.4% classified as high-risk », high-risk of what ? Please explain.

Response 3. The phrase means high-risk by MASCC score. It was added.

Point 4. Page 4, lines 163-165: Why were the risk factors recent hospitalization, recent antibiotics use, recent colonization with KPC-CPE, and recent ICU admission not included in Table 1 but kept separate in Table 2 ? These two tables should be fused as one.  Recent and previous colonization with KPC-CPE should also be defined in the table as well as recent antibiotic use. Additionally, the statistical tests used to obtain p-values in these tables should be given in the footnote.

Response 4. Done

Point 5. Page 5, Figures 1 & 2: the number of strains for each occurrence should be given along with the percentage. Indeed, from the materials & methods section, the reader understands that not all of the strains were screened for blaKPC and blaOXA-48 genes (Page 12, lines 458-460). Precise numbers should therefore be given to ascertain the relevance of these observations.

Response 5.

Figures 1 & 2: done

Materials & Methods: All isolates had resistant mechanisms confirmed by PCR (blaKPC and blaOXA-48), and 17 were studied with a multiplex PCR for other genes. It was added.

Point 6. Page 6: Figure 3 is unnecessary for understanding the paper and establishing the proposed score. It should therefore be removed.

Response 6. Done

Point 7. Page 6, table 3: there is room for improvement for this table. First, it should stand on a single page along with its footnote for a better understanding. Second, the item « clinical source » makes no sense to this reader as well as the statistical analysis held on these values. What do the authors mean with this item? Should it not simply be a category under which « Abdominal » « Central veinous catheter » and « Urinary » are found?

Response 7. We clarify the item as “bacteremia with clinical source” (not primary) followed by the main sources. The frequency of occurrence of these variables was compared between CSE and CRE.

Point 8.  Discussion: this section is well researched, with a number of studies addressing a similar problem described with their limitations. However, some sentences are misleading and should be toned down. For example, Page 9, lines 313-4, the authors state «Identifying patients at risk for CRE has a significant clinical impact since adequate empirical treatments can be implemented early, thus reducing mortality ». This statement is contradictory with the fact that recent antibitioc use was identified as a risk factor for CRE bacteremia in this study.

Response 8. We do not think the sentences are contradictory because receiving any antibiotic before a bacteremia episode is a risk factor for CRE. The score identifies patients at risk in order to prescribe an adequate empirical treatment to cover CRE.

Point 9. Discussion. In this reader’s point of view, the main interest of a score such as the one established in this work is to avoid unnecessary antibiotic treatment in patients identified as low risk and the emphasis should be put on this item rather that adequate empirical treatment for CRE.

Response 9. We agree with the reviewer. The most relevant issue of the score was emphasized.

Point 10. Discussion. Moreover, the most consistent risk factor for CRE bacteremia identified in this work and others is the colonization with CRE (mainly intestinal carriage identified through weekly rectal swabs). In such patients, the possibility to reduce the risk through Fecal Microbial Transplantation (FMT) rather than antibiotic use should therefore also be discussed.

Response 10. We agree with the reviewer. A comment on this subject was added.

Point 11. Page 11, line 407: « Patients already enrolled in the study…were excluded from the analysis ». This statement is puzzling; please clarify.

Response 11: it was removed

Point 12. From page 3, please use K. pneumoniae consistently throughout the manuscript

Response 12: done

Point 13.  Page 3, line 118 : please define ROCAS in « ROCAS study »

Response 13: done

Point 14. Page 3, line 155 : what kind of solid tumors were included and how many of each type ?

Response 14: It was answered in response 2

Point 15. Pages 5 & 6 : figure titles should be positionned below the figures and not above.

Response 15: done

Point 16. Page 5, figure 2 : ESBL abbreviation should also be explained in the caption

Response 16: done

Point 17. References : please check the correct use of italics for species names throughout this section

Response 17: done

Round 2

Reviewer 2 Report

Authors have made significant improvements in recommended styling and length of sections.

Methodologically, authors' reply has shown adequate methodological knowledge and justified approaches employed for risk scoring while addressing possible limitations. 

No further comments.